# Integrative Analysis of Gene Networks Associated with Adipose and Muscle Traits in Hanwoo Steers

**DOI:** 10.3390/ani15213201

**Published:** 2025-11-03

**Authors:** Suk Hwang, Taejoon Jeong, Junyoung Lee, Woncheoul Park, Sunsik Jang, Dajeong Lim

**Affiliations:** 1Department of Animal Resources Science, College of Agriculture and Life Sciences, Chungnam National University, Daejeon 34134, Republic of Korea; hwangseok1025@gmail.com (S.H.); jungtaejoon@gmail.com (T.J.); dlwnsdud0714@naver.com (J.L.); 2National Institute of Animal Science, Wanju 55365, Republic of Korea; wcpark1982@korea.kr (W.P.); jangsc@korea.kr (S.J.)

**Keywords:** fat tissue, Hanwoo, hub genes, muscle tissue, RNA-Seq, WGCNA

## Abstract

**Simple Summary:**

Hanwoo cattle are valued for their excellent meat quality and distinctive marbling; however, the genetic reasons behind the development of fat and muscle remain unclear. In this study, we analyzed gene expression in five tissues to understand how certain genes interact to shape these traits. We identified several groups of genes that were more active either in fat or in muscle. In fat tissue, many genes were related to lipid metabolism and tissue structure, whereas in muscle tissue, genes were primarily involved in energy production and mitochondrial function. Some genes, such as *AGPAT5* and *ARPC5*, seemed to play key roles in linking these biological processes. These results give us a clearer picture of how gene networks contribute to the balance between fat and muscle in Hanwoo cattle. The findings may help improve breeding programs by enhancing marbling and reducing excess fat, ultimately supporting better beef quality and sustainable production.

**Abstract:**

This study aims to characterize tissue-specific expression patterns in Hanwoo steers by identifying co-expression modules, functional pathways, and hub genes related to fat and muscle traits using Weighted Gene Co-expression Network analysis (WGCNA). RNA-Seq data were generated from three muscle tissues (longissimus muscle, tenderloin, and rump) and two fat tissues (back fat and abdominal fat) collected from six 30-month-old Hanwoo steers. Quality control of raw sequencing reads was performed using FastQC, and trimmed reads were aligned to the bovine reference genome (ARS-UCD1.3) using HISAT2. We also identified a gene co-expression network via *WGCNA* using normalized gene expression values. Modules were defined based on topological overlap and correlated with tissue-specific expression patterns. Modules with a significant association (*p* < 0.05) were used for functional enrichment based on Gene Ontology (GO) and KEGG pathways, as well as Protein–Protein Interaction Network analysis. A total of seven co-expression modules were identified by WGCNA and labeled in distinct colors (yellow, blue, red, brown, turquoise, green, black). Among them, the yellow and blue modules were positively associated with back fat, while the turquoise and green modules showed a negative correlation with abdominal fat. Additionally, the turquoise or green module was positively correlated with longissimus and rump tissues, indicating distinct gene expression patterns between fat and muscle. This study identified key co-expression modules and hub genes associated with muscle and fat metabolism. Notably, *ARPC5* (blue module) was involved in lipid metabolism and energy storage, whereas *AGPAT5* (turquoise module) was linked to maintaining muscle cell structure and function. These findings reveal biological mechanisms for tissue-specific gene regulation, providing targets for enhancing meat quality in Hanwoo.

## 1. Introduction

The accumulation and remodeling of adipose and muscle tissues are fundamental physiological processes that directly impact carcass traits and beef quality. In livestock species, the composition and distribution of fat and muscle tissues are critical determinants of meat quality, yield grade, and market value. Beef quality is primarily shaped during the fattening period, a critical phase in cattle production when rapid accumulation of intramuscular and subcutaneous fat occurs. During this stage, key carcass traits such as marbling, back fat thickness, meat color, and texture are largely established. These traits are influenced by a combination of genetic factors, nutrition, age, and sex. However, the extent of fat deposition, particularly intramuscular fat, is considered a major determinant of eating quality, including tenderness, juiciness, and flavor, especially in premium beef markets such as Korea [1]. Hanwoo (Korean native cattle) is an indigenous breed in Korea, highly valued for its exceptional intramuscular fat (marbling) and meat tenderness. In the Korean beef production system, Hanwoo steers are typically raised for approximately 30 months before slaughter, during which they undergo a distinct late fattening stage. This period is characterized by rapid deposition of both subcutaneous and intramuscular fat, making traits such as back fat thickness and marbling score economically important. These traits are regulated by coordinated biological mechanisms, including lipid metabolism, extracellular matrix (ECM) remodeling, and mitochondrial function [2]. Understanding how these processes are regulated at the transcriptional level in fat and muscle tissues is essential for advancing precision breeding strategies and improving meat quality.

RNA sequencing (RNA-Seq) enables comprehensive analysis of gene expression across diverse tissues, yet traditional differential expression approaches often fail to capture complex gene–gene interaction networks underlying tissue-specific functions. To address this, Weighted Gene Co-expression Network Analysis (WGCNA) provides a powerful systems biology framework for identifying co-expressed gene modules and detecting key regulatory hub genes [3]. WGCNA constructs networks based on pairwise expression correlations and groups genes into modules according to their co-expression patterns. This enables the discovery of central genes regulating entire modules, rather than focusing on single gene effects. Such a network-based approach is especially valuable in livestock, where complex traits like marbling or muscle growth are governed by multiple interconnected genes. While WGCNA has been applied to investigate adipogenesis and muscle development in cattle, few studies have directly compared fat and muscle tissues in Hanwoo steers at the terminal fattening stage. However, adipose and muscle tissues are controlled by distinct gene expression programs, which are tissue-specific regulatory patterns that govern cellular function. As a result, they contribute differently to carcass traits. Comparing their transcriptional networks can reveal how such mechanisms diverge between fat and muscle. Several crucial genes associated with back fat thickness in beef cattle—such as *ACACA*, *SCD*, *FASN*, *ACOX1*, *ELOVL5*, *HACD2*, and *HSD17B12*—have been identified through gene co-expression networks and linked to lipid metabolism, suggesting their roles as key regulators of fat deposition [4]. These findings highlight the utility of network-based approaches in uncovering coordinated regulatory programs underlying complex traits like fat deposition. Furthermore, integrating WGCNA-derived network centrality with protein–protein interaction (PPI) analysis allows for a more stringent identification of biologically relevant hub genes, which can serve as potential biomarkers or targets for selective breeding.

Therefore, we employed WGCNA to systematically explore transcriptional patterns across fat and muscle tissues in Hanwoo steers. In this study, we analyzed RNA-Seq data from five tissues: three muscle tissues (longissimus, tenderloin, and rump) and two fat tissues (abdominal and back fat)-collected from six 30-month-old Hanwoo steers. We constructed gene co-expression networks and identified gene co-expression modules associated with fat and muscle traits. Finally, PPI analysis was conducted to identify true hub genes that may have potential regulatory importance. Functional enrichment analysis was also performed to characterize the biological roles of each module. Our findings provide new insights into the transcriptional architecture of fat and muscle development in Hanwoo cattle and suggest candidate genes for improving beef quality through molecular breeding approaches.

## 2. Materials and Methods

### 2.1. Sample Collection and RNA-Seq Data Generation

To perform Weighted Gene Co-expression Network Analysis (WGCNA) in Hanwoo cattle, six 30-month-old steers from the same farm were obtained from the Hanwoo Cattle Research Institute, National Institute of Animal Science, Republic of Korea. The six samples were collected at the time of slaughter. To investigate gene expression through RNA-Seq, we collected five tissue types from six 30-month-old Hanwoo steers: three from muscle (*longissimus* muscle [sirloin/LOM], tenderloin [TEN], and round [RMP]) and two from fat tissues (back fat [BFT] and abdominal fat [ABF]).

### 2.2. Quality Control and Mapping of RNA-Seq Data

RNA-Seq libraries were prepared from three muscle tissues (longissimus muscle, tenderloin, and rump) and two fat tissues (back fat and abdominal fat) dissected from six 30-month-old Hanwoo steers. Raw paired-end reads (n = 29) underwent initial quality assessment with FastQC (fastqc v0.12.1), after which adapter sequences and low-quality bases were removed using Trimmomatic (v0.39) [5] in paired-end mode. Specifically, the TruSeq3-PE adapter file was applied with a seed mismatch threshold of 2, palindrome clip threshold of 30, and simple clip threshold of 10; bases with Phred scores < 3 were trimmed from both read ends, a 4-bp sliding window requiring a minimum average quality of 15 was enforced, and reads shorter than 36 bp were discarded. Clean reads were aligned to the reference genome (Bos taurus ARS-UCD1.3) using HISAT2 (version 2.2.1) [6]. Gene-level quantification was performed using featureCounts (v2.1.1) [7].

### 2.3. Construction of Gene Co-Expression Network of Fat and Muscle Tissues

WGCNA (v1.73) was used to construct gene co-expression networks based on variance-stabilized normalized gene expression values obtained from DESeq2. A soft-thresholding power of 16 was selected based on the criterion that the scale-free topology model fit approached saturation (R^2^ ≈ 0.9). Using this threshold, we constructed a signed network and identified a total of seven co-expression modules, excluding the gray module, which contains unassigned genes. We generated a heatmap of module eigengenes to evaluate the correlations between modules using the R package (CorLevelPlot 0.99.0). This allowed us to assess whether the modules exhibited positive or negative relationships visually. Modules were interpreted based on the significance of correlation (*p*-value < 0.01), and subsequent biological insights were inferred accordingly.

### 2.4. Identification of Co-Expressed Gene Modules in Fat and Muscle Tissues via WGCNA

In gene co-expression networks, hub genes are those that exhibit the highest level of connectivity within their respective modules, often acting as key regulators of the module’s biological functions. To identify such genes, we computed module membership scores by correlating each gene’s expression pattern with its corresponding module eigengene using Pearson correlation. We employed the chooseTopHubInEachModule() function of WGCNA and then selected genes with the highest intramodular connectivity for each module.

### 2.5. Functional Enrichment Analysis of Tissue-Specific Gene Co-Expression Modules

Functional enrichment analysis was conducted on the seven modules using the clusterProfiler R package (clusterProfiler_4.14.6) [8]. Significant Gene Ontology (GO) [9] terms-Biological Process (BP), Molecular Function (MF), and Cellular Component (CC)-as well as KEGG pathways [10] were identified to elucidate the biological functions of each module.

### 2.6. Identification of Module-Specific Hub Genes via PPI and Functional Network Analysis

We can explore the PPI networks of key genes within each module using the STRING database (v12.0, https://string-db.org, accessed on 15 August 2025). In addition, the CytoHubba plugin (v0.1) was applied to evaluate the network topology [11] of the PPI network. Maximal Clique Centrality (MCC) was used to rank genes based on their topological importance within the network. Therefore, we identified hub genes by ranking nodes according to Maximal Clique Centrality alongside other network topology metrics, selecting the top 30 most highly connected nodes as our final hub genes [11].

We constructed and visualized functional networks of enriched KEGG pathways using the Cytoscape plugins ClueGO (v2.5.10) [12] and CluePedia (v1.5.10) [13]. Enrichment analysis was conducted on genes from tissue-specific co-expression modules identified via WGCNA. We used the default parameters provided by ClueGO and CluePedia, in which functionally related terms were grouped using kappa statistics (≥0.4), and pathways with *p*-values < 0.05 were considered significantly enriched. The hypergeometric test and Benjamini–Hochberg correction were applied for statistical evaluation.

## 3. Results

### 3.1. QC and Mapping of RNA-Seq Data

The average number of input reads across samples was approximately 36.97 million. On average, 31.86 million reads were uniquely aligned, while 1.98 million were aligned to multiple locations. Roughly 3.13 million reads per sample remained unaligned. The mean overall alignment rate was 98.2%, indicating high-quality and consistent mapping across all samples.

### 3.2. Construction of Gene Co-Expression Network of Fat and Muscle Tissues

We used WGCNA to analyze gene co-expression patterns associated with fat and muscle tissues from six 30-month-old Hanwoo steers using normalized gene expression values. From a total of 31,841 genes, 14,017 were retained for downstream analysis after filtering out genes with fewer than 15 read counts in at least 24 out of 30 samples (≥80%) based on RNA-Seq data from 24 tissue samples. Before network construction, we assessed whether transcriptomic variation reflected tissue origin. Principal component analysis (PCA) was performed based on variance-stabilized transformed (*vst*) expression values. As shown in Figure 1, the samples clustered clearly according to tissue type, with fat-derived tissues forming a distinct cluster on the left and muscle-derived tissues clustering on the right. This separation indicates that tissue-specific transcriptional signatures are strongly preserved, with minimal overlap between fat and muscle samples. The first principal component (PC1) accounted for 93% of the total variance, while PC2 explained 1%, demonstrating that PC1 primarily captures variation driven by tissue-specific gene expression. To determine the optimal soft-thresholding power (β), the pickSoftThreshold() function was applied, and a power of 16 was selected based on the scale-free topology criterion (signed R^2^ > 0.9) (Figure 2A,B). A signed co-expression network was then constructed using the blockwiseModules() function. This procedure identified a total of eight co-expression modules, each labeled by a distinct color (excluding the gray module, which represents unassigned genes). The number of genes in each module varied widely, ranging from 55 genes in the black module to 5987 genes in the turquoise module. Among the identified modules, turquoise (5987 genes), blue (3501 genes), and brown (280 genes) encompassed the majority of annotated genes. In contrast, smaller modules—including yellow (170 genes), green (145), red (79), and black (55)—were not excluded, as they showed potential for tissue-specific relevance and were thus included in subsequent analyses. Module eigengenes (the first principal components of each module) were calculated and correlated with five tissues. The module–trait correlation heatmap revealed that the blue module exhibited a strong positive correlation with abdominal fat. In contrast, the turquoise module was closely associated with muscle-enriched tissues, including longissimus and tenderloin. These modules were selected for downstream functional enrichment analysis and identification of hub candidate genes.

To investigate biologically meaningful associations between gene co-expression modules and tissue traits, we calculated Pearson correlation coefficients between module eigengenes and five tissue traits. The module–trait correlation heatmap can identify modules that are strongly associated with fat- or muscle-related characteristics (Figure 3). The blue module was significantly positively correlated with abdominal fat (r = 0.69, *p* < 0.001) and back fat (r = 0.48, *p* < 0.01), while showing significant negative correlations with all muscle tissues, including longissimus (r = −0.40, *p* < 0.05), rump (r = −0.39, *p* < 0.05), and tenderloin (r = −0.38, *p* < 0.05), indicating a strong adipose-specific transcriptional signature and muscle-depleted expression. In contrast, the turquoise module showed a significant negative correlation with abdominal fat (r = −0.54, *p* < 0.01) and back fat (r = −0.65, *p* < 0.001), while exhibiting a positive correlation with longissimus (r = 0.37, *p* < 0.05), rump (r = 0.36), and tenderloin (r = 0.46, *p* < 0.01), suggesting enrichment in muscle-specific gene expression and depletion in adipose-related transcription. The yellow module displayed the strongest positive correlation with back fat (r = 0.73, *p* < 0.001), suggesting involvement in back fat–specific gene regulatory processes; it also showed a weak positive correlation with abdominal fat (r = 0.21) and a significant negative correlation with tenderloin (r = −0.42, *p* < 0.05), reflecting partial adipose-specific and tenderloin-depleted expression. Lastly, the green module was strongly negatively correlated with abdominal fat (r = −0.58, *p* < 0.001) and positively correlated with longissimus (r = 0.47, *p* < 0.01) and rump (r = 0.42, *p* < 0.05), suggesting that it captures muscle-enriched and adipose-depleted transcriptional signals. Among these, the yellow and blue modules showed positive associations with back fat, while the turquoise and green modules were negatively correlated with abdominal fat.

### 3.3. Identification of Hub Genes in Fat and Muscle Tissues via WGCNA

Based on intramodular connectivity calculated using the Topological Overlap Matrix (TOM), the hub gene of each co-expression module was identified as the gene with the highest topological connectivity. As summarized in Table 1, the top hub genes identified for each module were as follows: *DDR2* (yellow), *AGPAT5* (turquoise), *RREB1* (brown), *RPL34* (red), *ARPC5* (blue), *LOC112442783* (black), and *LOC112443631* (green). These genes represent the most interconnected nodes within their respective modules and were selected for downstream biological interpretation and network visualization.

### 3.4. Functional Enrichment Analysis in Each Module Using Gene Ontology (GO) and KEGG Pathway

As shown in Figure 4, the blue module exhibited strong positive correlations with adipose tissues, particularly abdominal and back fat, and was functionally enriched in biological processes related to lipid metabolism and cytoskeletal organization. Key GO-Biological Process (GO-BP) terms included *actin filament-based process* (GO:0030029, adj. *p* = 9.69 × 10^−6^), *lipid biosynthetic process* (GO:0008610, adj. *p* = 8.33 × 10^−6^), and *vesicle organization* (GO:0016050, adj. *p* = 1.65 × 10^−4^), suggesting active cytoskeletal remodeling and lipid handling in adipose tissues. In contrast, the turquoise module positively correlated with skeletal muscle tissues, was enriched for mitochondrial functions, and protein degradation pathways. Significant GO-BP terms included *protein catabolic process* (GO:0030163, adj. *p* = 2.0 × 10^−14^), *mitochondrion organization* (GO:0007005, adj. *p* = 5.5 × 10^−14^), and *cellular respiration* (GO:0045333, adj. *p* = 6.0 × 10^−14^), indicating elevated metabolic activity in oxidative muscle fibers. Consistent with these findings, KEGG pathway analysis of the turquoise module revealed strong enrichment in *oxidative phosphorylation* (bta00190, adj. *p* = 1.52 × 10^−29^), *thermogenesis* (bta04714, adj. *p* = 6.74 × 10^−16^), and *Parkinson disease* (bta05012, adj. *p* = 1.01 × 10^−15^), all of which are related to mitochondrial energy metabolism and cellular stress responses (Figure 5). Although the yellow module showed the strongest positive correlation with back fat in the module–trait relationship analysis, its functional enrichment results were relatively modest. GO analysis identified a small number of significantly enriched structural terms, including *extracellular matrix* (GO:0031012, adj. *p* = 9.24 × 10^−3^), *external encapsulating structure* (GO:0030312, adj. *p* = 9.24 × 10^−3^), and *collagen-containing extracellular matrix* (GO:0062023, adj. *p* = 9.24 × 10^−3^). These findings suggest that the yellow module may play a role in extracellular matrix remodeling associated with adipose tissue expansion.

### 3.5. Identification of the Trait-Associated Hub Genes from Gene Modules

To identify core regulatory genes within key tissue-specific modules, we applied an integrated strategy combining co-expression network centrality and protein–protein interaction (PPI) network analysis. Candidate hub genes were defined by three stringent criteria: (i) high intramodular connectivity (module membership, MM > 0.9), (ii) top-ranked PPI network centrality, and (iii) exclusion of housekeeping (HK) genes to ensure tissue specificity. The true hub genes identified from each module are visualized in Figure 6; red-circled nodes denote genes that met all selection criteria. Although prior studies have considered MM > 0.8 as a valid threshold to identify potential hub genes within WGCNA modules [14], we adopted a more conservative cutoff of MM > 0.9 to enhance specificity and biological robustness in hub gene selection. This stricter criterion was particularly important given the high dimensionality of transcriptomic data and the need to identify strongly co-regulated, tissue-specific drivers relevant to adipose biology in the late fattening stage of Hanwoo steers. The Blue module was previously shown to be significantly positively correlated with abdominal and back fat tissues. A total of 3501 genes were assigned to this module. Among these, 568 genes satisfied MM > 0.9. The PPI network analysis identified 30 genes with the highest centrality scores. Cross-referencing these two sets and removing HK genes resulted in nine candidate hub genes: *FCGR3A*, *ANXA5*, *CD163*, *MYD88*, *HGF*, *TLR2*, *CD86*, *TLR4*, and *CD44*. These genes exhibited both strong co-expression with the module eigengene (MM > 0.9) and high centrality within the PPI network. *TLR2*, *TLR4*, *MYD88*, and *CD86* genes are integral components of inflammatory signaling pathways. In bovine adipose tissue, *TLR4* expression correlates with inflammatory cytokines and lipid release mechanisms [15], and non-esterified fatty acids can activate TLR4-mediated NF-κB signaling, promoting metabolic dysregulation in adipose cells [16]. Additionally, opposing yet regulatory roles of TLR2 and TLR4 in adipose tissue homeostasis have been demonstrated [17], highlighting their roles in immune-metabolic crosstalk. HGF and CD44 are also known to modulate adipocyte proliferation and extracellular matrix interactions, indicating their potential regulatory role in fat expansion or remodeling [18]. The yellow module was strongly associated with back fat-specific expression and extracellular matrix remodeling. Among the total of 170 genes, 21 genes surpassed the MM > 0.9 threshold. By integrating this list with the top 30 PPI-ranked genes and excluding known housekeeping genes, a refined set of nine true hub genes was obtained: *FBN1*, *MFAP5*, *MFAP2*, *LOX*, *TNXB*, *MMP2*, *COL14A1*, *ITGBL1*, and *CD34*. These genes are functionally linked to ECM architecture, collagen cross-linking, and cell adhesion, supporting a tightly coordinated regulation of tissue remodeling processes within adipose depots.

In contrast, the turquoise module, which comprised 5987 genes, showed strong positive correlations with muscle traits (longissimus and tenderloin). Of these, 487 genes met the MM threshold of greater than 0.9. PPI centrality analysis identified a distinct group of mitochondrial ribosomal proteins. After filtering for MM > 0.9 and excluding HK genes, we identified 24 true hub genes: *MRPL16*, *MRPL19*, *MRPL24*, *MRPL27*, *MRPL30*, *MRPL35*, *MRPL45*, *MRPL47*, *MRPL48*, *MRPL51*, *MRPL55*, *MRPL9*, *MRPL13*, *MRPS7*, *MRPS11*, *MRPS14*, *MRPS15*, *MRPS16*, *MRPS18C*, *MRPS25*, *MRPS30*, *PTCD3*, *MRRF*, *GADD45GIP1*. These genes are heavily involved in mitochondrial translation and energy metabolism, suggesting a central role for mitochondrial function in skeletal muscle regulation in Hanwoo. The summary of gene counts, PPI candidates, and final true hub genes for each module is provided in Table 2.

A total of 9, 24, and 9 true hub genes were identified in the blue, turquoise, and yellow modules, respectively, based on high module membership (MM > 0.9) and PPI network centrality (top 30).

### 3.6. Functional Interpretation of Tissue-Specific Gene Modules Using ClueGO and CluePedia

Importantly, many of the true hub genes identified based on both high WGCNA module membership (MM > 0.9) and PPI network centrality were closely associated with the key biological processes revealed through ClueGO and CluePedia. In the blue module, significant enrichment was observed for immune-associated processes, such as the response to lipoprotein particles, tuberculosis signaling, and leukocyte chemotaxis. The five hub genes—*TLR2*, *TLR4*, *MYD88*, *CD86*, and *FCGR3A*—are clustered within innate immune signaling pathways, indicating their potential role in inflammation and host defense, particularly in response to lipid stimuli (Figure 7A). In contrast, the turquoise module was enriched for mitochondrial energy-related pathways, including the respiratory chain and oxidative phosphorylation. Hub genes such as *NDUFS3*, *SDHB*, *UQCRC1*, and *COX7C* occupied central positions in the functional network, underscoring their essential roles in ATP production and supporting the turquoise module’s identity as a regulator of muscle metabolism (Figure 7B). Meanwhile, the yellow module was primarily linked to extracellular matrix (ECM) structure and developmental processes, with functional terms such as collagen fibril organization, embryo implantation, and ECM assembly significantly overrepresented. Corresponding hub genes—including *COL1A2*, *COL6A3*, *TNXB*, and *PCOLCE2*—were embedded in collagen-related clusters within the ClueGO map, emphasizing their role in ECM remodeling (Figure 7C). Together, the alignment between each module’s biological function and the network positions of its top hub genes illustrates a robust tissue-specific regulatory framework that likely plays a pivotal role in orchestrating fat and muscle trait development in Hanwoo cattle. The complete edge list containing all gene–gene interaction pairs (MCC top-30 with shortest paths) from the blue, yellow, and turquoise modules is provided as Appendix A. This file lists pairwise gene–gene connections (source and target) used to construct the Cytoscape networks, enabling full reproduction of the module-level co-expression topology.

## 4. Discussion

To explore the regulatory mechanisms underlying fat and muscle development in Hanwoo steers, we constructed a gene co-expression network and identified trait-associated modules using WGCNA. Of these, three modules—blue, yellow, and turquoise—were prioritized based on their strong trait correlations and biological relevance. Based on the WGCNA findings, this discussion highlights the key co-expression modules that represent tissue-specific regulatory mechanisms relevant to Hanwoo beef quality.

In the blue module, *ARPC5* (MM = 0.990448156) was identified as the most central hub gene, and functional enrichment analysis revealed significant overrepresentation of terms including actin cytoskeleton organization (GO:0030036, adj. *p* = 3.69 ×10^−9^) and actin filament-based process (GO:0030029, adj. *p* = 9.69 × 10^−9^). These biological processes are closely tied to cytoskeletal remodeling, a critical component of both preadipocyte differentiation and immune cell migration in adipose tissue.

Interestingly, *CD44* and *TLR4* also emerged as strong true hub gene candidates and are known to play essential roles in actin cytoskeleton dynamics and inflammation-mediated tissue remodeling. In bovine preadipocyte differentiation experiments, *CD44* was shown to directly regulate lipid droplet accumulation, triglyceride levels, and adipocyte-specific gene expression, including *LPL* and *FABP4*, demonstrating its critical role in adipogenesis in cattle [19]. These findings are consistent with previous observations in murine models, where a high-fat diet (HFD) induced *CD44* expression and promoted macrophage infiltration and metabolic dysfunction [20]. In bovine adipocytes, activation of *TLR4* by lipopolysaccharide (LPS) has been shown to trigger prostaglandin E_2_–E_2_-dependent lipolysis, highlighting its key role in linking inflammation and lipid mobilization during the fattening stage in cattle [21]. Given that *ARPC5* encodes a core subunit of the Arp2/3 complex essential for actin nucleation [22], these converging roles suggest the presence of a functional *CD44*–*TLR4*–*ARPC5* regulatory axis that integrates immune signaling with cytoskeletal remodeling during adipose tissue adaptation.

In addition to cytoskeletal pathways, enrichment analysis also uncovered metabolic signatures including lipid biosynthetic process (GO:0008610, adj. *p* = 8.33 × 10^−8^), fatty acid metabolism (KEGG: bta01212, adj. *p* = 1.00 × 10^−4^), and sphingolipid metabolic process (GO:0006665, adj. *p* = 1.65 × 10^−4^) suggesting that the blue module may coordinate lipid handling, membrane composition, and related signaling events during the late stages of fat development.

Considering that the animals analyzed were 30-month-old Hanwoo steers, a typical endpoint of the fattening phase when intramuscular and subcutaneous fat deposition intensifies [2], the biological and economic relevance of these findings is evident. At this stage, adipose tissue undergoes significant expansion, ECM remodeling, and metabolic shifts toward lipid accumulation and immune modulation. The convergence of key hub genes (*ARPC5*, *CD44*, *TLR4*) with both actin-related and lipid-related pathways reflects a sophisticated regulatory program that combines structural and immune mechanisms to influence carcass fat quality.

These findings are not isolated. Similar patterns involving the actin cytoskeleton and immune signaling pathways have been observed in other cattle breeds. For instance, Garcia et al. (2024) found that immune signaling and cytoskeletal genes, including TLR-family genes and actin-binding proteins, were upregulated during fattening in Nelore cattle [23], supporting the notion that the blue module reflects conserved molecular mechanisms of adipose remodeling.

Complementing the intracellular and immune-regulatory functions of the blue module, the yellow module, which also showed a strong positive correlation with back fat tissue, was enriched for extracellular matrix (ECM) remodeling. In the WGCNA, DDR2 (MM = 0.9738) was identified as a collagen-binding receptor tyrosine kinase activated by collagen. Few studies have shown that DDR2 is activated by fibrillar collagen and plays a key role in regulating fibroblast migration and proliferation, as well as extracellular matrix (ECM) remodeling [24].

Additionally, 9 true hub genes—*FBN1*, *MFAP5*, *MFAP2*, *LOX*, *TNXB*, *MMP2*, *COL14A1*, *ITGBL1*, and *CD34*—were selected based on high module membership (MM > 0.9) and PPI centrality, all of which are involved in ECM structure and adipose tissue support. GO enrichment further confirmed the module’s structural identity, with significant overrepresentation of ECM-related terms.

Although both the blue and yellow modules showed associations with adipose tissues based on the module–trait heatmap, the yellow module displayed a particularly strong and specific positive correlation with back fat thickness (r = 0.73, *p* < 0.001), while exhibiting only weak association with abdominal fat (r = 0.21) and a moderate negative correlation with muscle tissue such as tenderloin (r = −0.42). This pattern suggests that the yellow module may represent a back fat–dominant transcriptional network, potentially distinct from the more metabolically diverse or broadly adipogenic characteristics of the blue module.

This is especially relevant in Hanwoo beef production, where excessive back fat is considered undesirable due to its negative phenotypic correlation with intramuscular fat (IMF), a key trait for carcass quality and market value ([25,26]).

Despite its strong enrichment in ECM-related functions, the yellow module may reflect a maladaptive regulatory program associated with excessive subcutaneous fat deposition. Key hub genes such as *DDR2*, *FBN1*, and *MMP2* may play central roles in this process. Notably, similar findings have been reported in pigs, where ECM genes like COL14A1 and FN1 were linked to increased back fat thickness, supporting the idea that ECM remodeling contributes to subcutaneous fat accumulation [27].

However, *AGPAT5* (MM = 0.99421), a WGCNA-identified hub gene in the turquoise module, functions as a lysophosphatidic acid acyltransferase that catalyzes the conversion of lysophosphatidic acid to phosphatidic acid. Previous studies have demonstrated that *AGPAT5* is localized to the mitochondrial outer membrane, where it participates in lysophosphatidic acid acylation to generate phosphatidic acid [28]. Although direct functional studies of *AGPAT5* in cattle are limited, transcriptomic analyses in beef breeds have shown that lipid biosynthesis pathways—including *AGPAT* family genes—were upregulated in high-marbled intramuscular fat tissues, suggesting a potential role for *AGPAT5* in muscle-specific lipid remodeling and energy regulation during the late fattening phase [29].

This interpretation aligns well with the turquoise module’s expression profile, which showed a strong positive correlation with muscle traits and a negative association with back fat thickness. Enrichment analysis revealed strong links to mitochondria-centered metabolic pathways, including *oxidative phosphorylation* (KEGG: bta00190, adj. *p* = 8.31 ×10^−8^), *protein catabolic process* (GO:0030163, adj. *p* = 3.53 × 10^−21^), and *mitochondrial protein-containing complex* (GO:0098798, adj. *p* = 2.83 × 10^−32^), highlighting the module’s involvement in muscle energy metabolism and protein turnover.

We further identified 24 true hub genes within the turquoise module by applying strict selection criteria (MM > 0.9, top 30 PPI, excluding housekeeping genes). Most of these genes belong to the MRPL and MRPS families, nuclear-encoded components of the mitochondrial ribosome essential for mitochondrial protein synthesis and oxidative phosphorylation regulation [30]. Previous studies have shown that *MRPL* and *MRPS* genes are highly expressed in metabolically active tissues such as skeletal muscle and are associated with enhanced ATP production, feed efficiency, and mitochondrial biogenesis in cattle [31,32]. Their prominence suggests that the turquoise module encodes a transcriptional network geared toward supporting mitochondrial bioenergetics and growth in skeletal muscle. *AGPAT5*, while not a ribosomal gene, may complement this system by regulating mitochondrial membrane lipids and sustaining ATP production. These findings are consistent with previous studies in highly marbled cattle. Tegeler et al. (2025) reported that genes related to oxidative phosphorylation and mitochondrial function were significantly upregulated in the longissimus muscle of Japanese Black cattle with high intramuscular fat content [33], supporting the muscle-specific metabolic identity of the turquoise module.

In summary, the turquoise module displays strong biological characteristics associated with muscle-specific energy metabolism, particularly involving mitochondrial translation and protein turnover, while maintaining a clear inverse relationship with back fat thickness. This suggests that the module may support muscle development while at the same time limiting subcutaneous fat deposition. Such a dual regulatory effect is particularly relevant to premium Hanwoo beef production, where minimizing excessive back fat while enhancing intramuscular fat (marbling) remains a critical objective for improving meat quality. Altogether, the findings from the turquoise module provide new insights into the transcriptional programs regulating muscle growth and fat distribution during the late fattening stage. These results may contribute to identifying molecular targets for selective breeding and nutritional strategies to improve carcass quality in Hanwoo steers. Future studies incorporating larger and more diverse populations and functional validation experiments will be necessary to confirm the regulatory roles of these modules and their hub genes in muscle development. Collectively, this study provides the first integrative transcriptomic framework comparing fat and muscle tissues in Hanwoo cattle using WGCNA. Unlike previous studies that examined single tissues, our network-level approach uncovered cross-tissue regulatory modules controlling both fat deposition and muscle metabolism. The blue and yellow modules highlight structural and inflammatory processes shaping adipose expansion, while the turquoise module represents a metabolic network promoting muscle growth while suppressing excessive subcutaneous fat accumulation. 

These findings establish a molecular foundation for selective breeding programs aiming to enhance intramuscular fat (marbling) without increasing back fat thickness—a major goal in premium Hanwoo beef production. Moreover, the identified hub genes (e.g., *ARPC5*, *DDR2*, *AGPAT5*) can serve as candidate markers for genetic selection or nutritional modulation strategies to improve carcass quality and production efficiency in Korean native cattle.

## 5. Conclusions

In this study, we explored the gene expression networks underlying fat and muscle development in Hanwoo steers using a systems biology approach. By comparing five tissues from fat and muscle, we identified distinct gene modules that reflect their different biological roles. The blue module was strongly associated with adipose tissues and enriched for lipid metabolism and cytoskeletal organization, highlighting genes such as *ARPC5*, *CD44*, and *TLR4* that may link fat storage with immune and structural remodeling. In contrast, the turquoise module showed a clear association with muscle tissues and was enriched for mitochondrial function and energy metabolism, with *AGPAT5* and several mitochondrial ribosomal genes (*MRPL* and *MRPS* families) serving as key hub genes. The yellow module, correlated with back fat, was mainly involved in extracellular matrix remodeling through genes such as *DDR2* and *MMP2*. Together, these findings reveal coordinated genetic programs that distinguish fat from muscle functions during the late fattening phase in Hanwoo cattle. Understanding these gene networks provides valuable insights into the molecular basis of marbling and back fat development and offers potential targets for breeding strategies aimed at producing high-quality beef with improved balance between intramuscular and subcutaneous fat.

## Figures and Tables

**Figure 1 animals-15-03201-f001:**
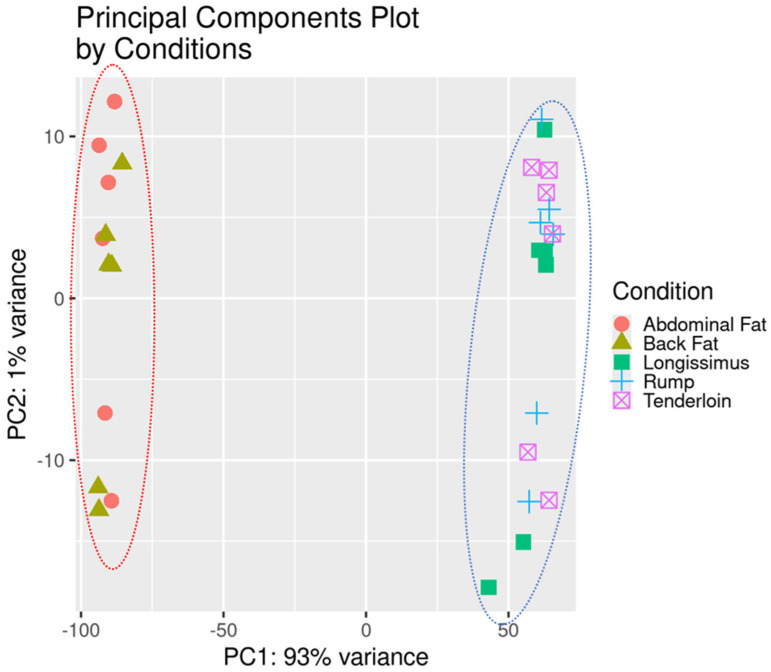
Principal component analysis (PCA) of gene expression profiles across five tissue types. PCA plot showing clear separation between adipose-derived (**left**) and muscle-derived (**right**) samples. Combined PCA plot integrating all tissues; PC1 and PC2 explained 93% and 1% of total variance, respectively.

**Figure 2 animals-15-03201-f002:**
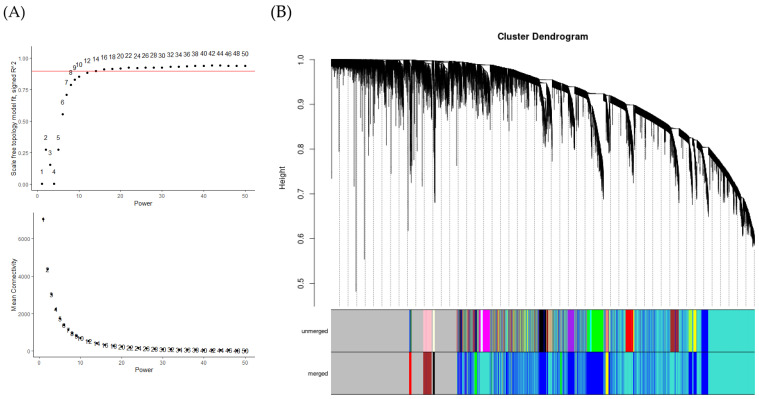
Weighted gene co-expression network analysis (WGCNA) of genes and identification of key modules. (**A**) Analysis of network topology for soft-thresholding powers for genes. When the correlation coefficient reached 0.9, the scale-free topology model fit approached its maximum value, and the optimal soft threshold was 16. The red line indicates the correlation coefficient threshold (0.9) used to determine the optimal soft-thresholding power. (**B**) Hierarchical clustering dendrogram showing module divisions; a total of eight color-coded modules were constructed.

**Figure 3 animals-15-03201-f003:**
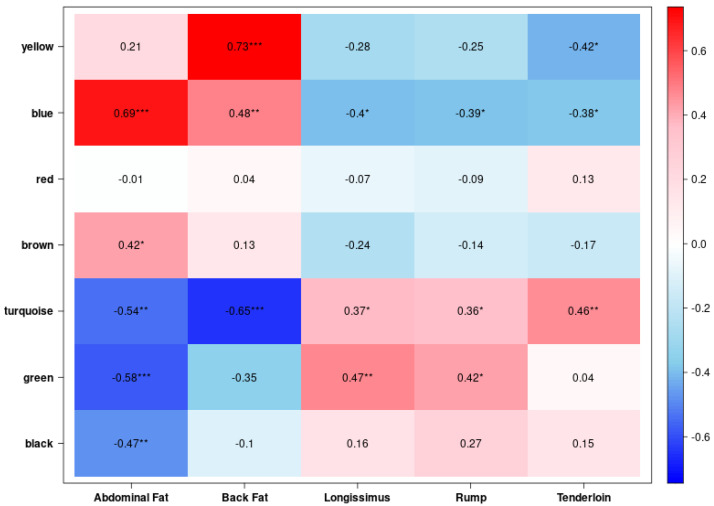
Weighted gene co-expression network analysis (WGCNA) of genes and identification of key modules. Heatmap showing correlations between seven gene modules and five tissue types. The yellow module exhibited the strongest positive correlation with back fat (0.73, ***), whereas the blue module correlated positively with abdominal fat (0.69, ***). The turquoise module displayed negative correlations with abdominal fat (−0.54, **) and back fat (−0.65, ***). The green module was negatively correlated with abdominal fat (−0.58, ***). Overall, yellow and blue modules were associated with fat accumulation, whereas turquoise and green modules showed inverse relationships with fat but positive associations with muscle traits. (* *p* < 0.05, ** *p* < 0.01, *** *p* < 0.001).

**Figure 4 animals-15-03201-f004:**
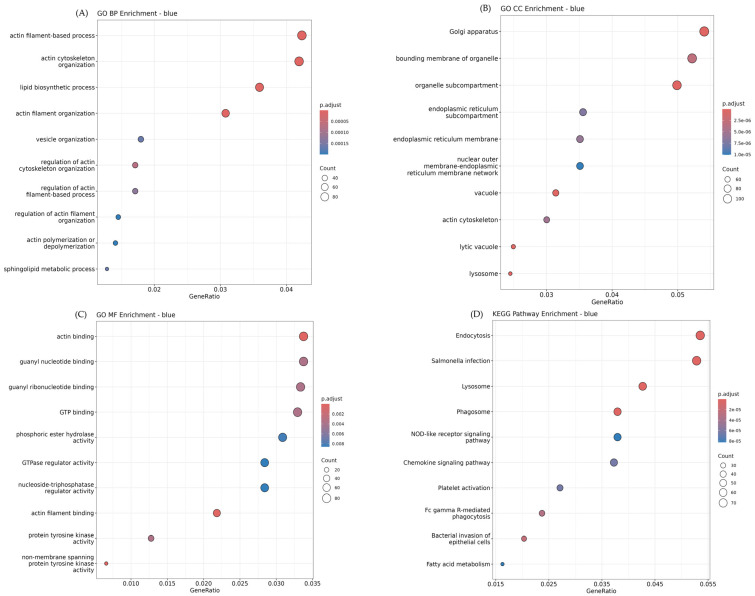
Functional enrichment results of the blue module. (**A**) GO Biological Process (BP): actin filament-based processes, lipid biosynthesis, and vesicle organization were enriched. (**B**) GO Cellular Component (CC): Golgi apparatus, endoplasmic reticulum membrane, and lysosome enrichment. (**C**) GO Molecular Function (MF): actin binding, GTP binding, and tyrosine kinase activity. (**D**) KEGG pathway analysis: endocytosis, phagosome, and fatty acid metabolism were significantly enriched.

**Figure 5 animals-15-03201-f005:**
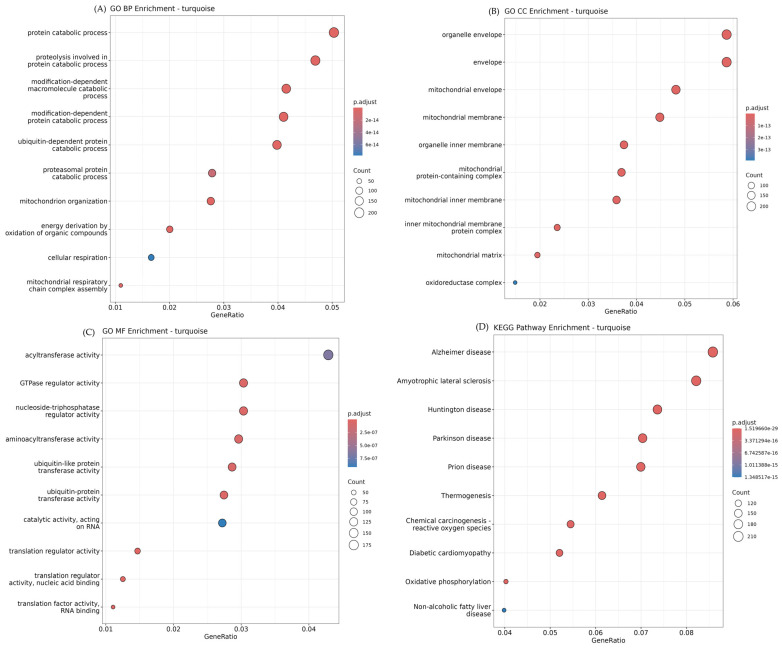
Functional enrichment results of the turquoise module. (**A**) GO Biological Process (BP): protein catabolism, mitochondrial organization, and cellular respiration were highly enriched. (**B**) GO Cellular Component (CC): enrichment in mitochondrial structures, including the inner membrane and the oxidoreductase complex. (**C**) GO Molecular Function (MF): translational regulation and acyltransferase activity. (**D**) KEGG pathway analysis revealed that oxidative phosphorylation, thermogenesis, and neurodegenerative disease pathways were the most dominant.

**Figure 6 animals-15-03201-f006:**
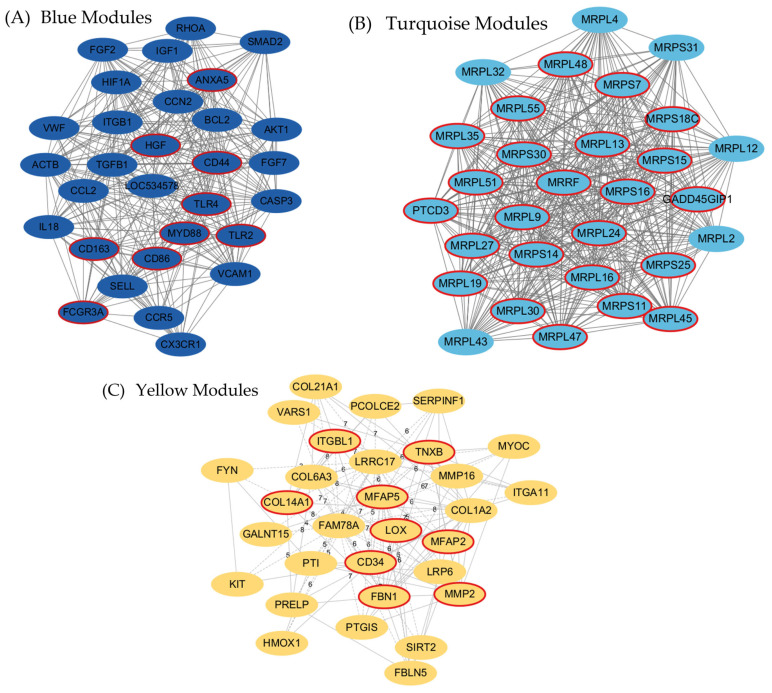
Identification of true hub genes through co-expression and PPI network integration. (**A**) Blue module (fat-specific): inflammation-related and membrane-interacting genes dominate the network. (**B**) Turquoise module (muscle-specific): mitochondrial ribosomal proteins form a densely connected cluster, indicating enhanced metabolic activity. (**C**) Yellow module (ECM-related): structural ECM regulators are highlighted as hub genes. Nodes with red circular outlines represent the final true hub genes, selected based on high intramodular connectivity (MM > 0.9), top PPI centrality, and exclusion of housekeeping genes.

**Figure 7 animals-15-03201-f007:**
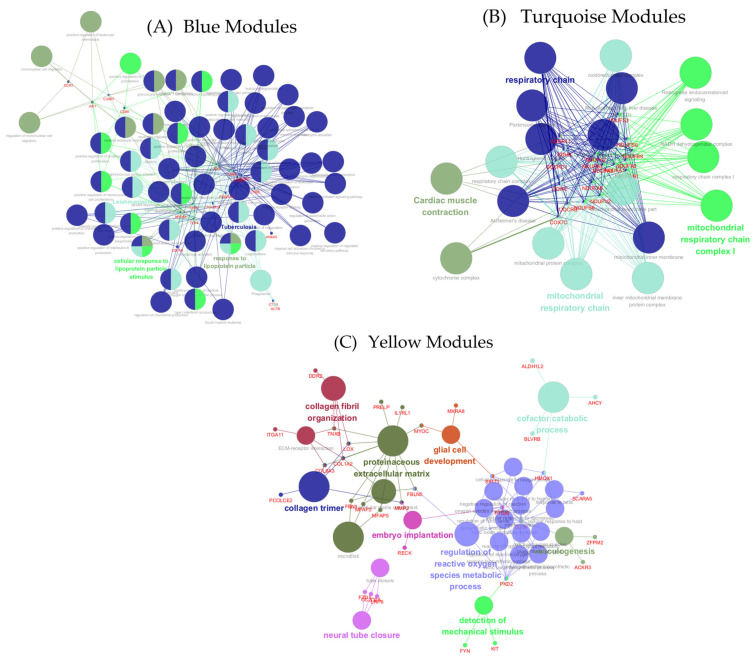
Functional enrichment networks of key WGCNA modules were constructed using ClueGO and CluePedia. (**A**) Blue module: innate immune and inflammatory signaling pathways. (**B**) Turquoise module: mitochondrial respiratory chain and oxidative phosphorylation pathways. (**C**) Yellow module: extracellular matrix organization and collagen-related processes. Hub genes are positioned within key regulatory clusters in each network.

**Table 1 animals-15-03201-t001:** Summary of the hub genes in the identified co-expression modules.

Module	Gene Count in Module	Hub Gene
Yellow	170	*DDR2*
Turquoise	5987	*AGPAT5*
Brown	280	*RREB1*
Red	79	*RPL34*
Blue	3501	*ARPC5*
Black	55	*LOC112442783*
Green	145	*LOC112443631*

*DDR2* (yellow), *AGPAT5* (turquoise), *RREB1* (brown), *RPL34* (red), *ARPC5* (blue), *LOC112442783* (black), and *LOC112443631* (green) represent the most interconnected nodes within their respective modules and were selected for biological interpretation and visualization.

**Table 2 animals-15-03201-t002:** Summary of true hub gene selection across key co-expression modules.

Module	Total Genes in Module	Genes MM > 0.9	PPI Top 30	True Hub Genes
Blue	3501	568	30	9
Turquoise	5987	487	30	24
Yellow	170	21	30	9

## Data Availability

Upon reasonable request, the datasets of this study can be made available from the corresponding author.

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
