# Peer review of "Integrative Analysis of Gene Networks Associated with Adipose and Muscle Traits in Hanwoo Steers"

_animals, 2025, doi:10.3390/ani15213201_

Round 1

Reviewer 1 Report

Comments and Suggestions for Authors

The manuscript “Integrative analysis of gene networks associated with adipose and muscle traits in Hanwoo steers” applies RNA-Seq and WGCNA to identify tissue-specific gene modules and hub genes related to fat and muscle traits in Hanwoo cattle. The study is methodologically sound, results are clearly presented, and findings are relevant for improving beef quality and selective breeding. However, several minor issues should be addressed before publication.

Suggestions for improvement

Clarify the inconsistency in Section 2.1 regarding animal age (stated as 30 months but reported as “Mean ± SD, 15.6 ± 5.5”).

Explicitly acknowledge the limitation of the small sample size (six animals) on generalizability.

Revise the Ethics Approval section to reflect that approval was obtained (NIAS20201979), instead of “Not applicable.”

Improve readability of dense figures (e.g., PPI networks) or move detailed networks to Supplementary Materials.

Streamline the discussion to avoid repetition and emphasize novelty and practical implications for Hanwoo breeding.

Perform careful proofreading to correct minor grammatical issues and redundancies.

Important: Please verify that the manuscript strictly follows the official MDPI template and formatting guidelines before final submission.

Author Response

Comment 1: Clarify the inconsistency in Section 2.1 regarding animal age (stated as 30 months but reported as “Mean ± SD, 15.6 ± 5.5”).
Answer: We appreciate the reviewer’s observation. The value “15.6 ± 5.5” was mistakenly retained from an earlier draft and has been removed. The correct age (30 months) is now consistently described throughout the manuscript. (Section 2.1, Sample collection and RNA-Seq data generation).

Comment 2: Explicitly acknowledge the limitation of the small sample size (six animals) on generalizability.
Answer: We initially collected samples from 10 Hanwoo steers; however, only six animals had all tissue types (rump, longissimus, tenderloin, abdominal fat, and back fat) available for analysis. Therefore, these six animals were used for the integrated analysis. We acknowledge that the limited sample size may reduce statistical power; however, the inclusion criteria were necessary to ensure data consistency and comparability across tissues. Moreover, similar livestock transcriptomic experiments use three or more biological replicates — for instance, bovine adipose RNA-seq with three biological replicates per breed (Song et al., 2019) and whole-blood RNA-seq with six dairy bulls (Zhao et al., 2023) — indicating that n = 3–6 biological replicates is common practice in livestock transcriptomics. These precedents support the appropriateness of our n = 6 design for integrated, cross-tissue analyses.

Comment 3: Revise the Ethics Approval section to reflect that approval was obtained (NIAS20201979), instead of “Not applicable.”
Answer: Corrected. The Ethics Approval statement now reads:

“Ethics approval was obtained from the National Institute of Animal Science (approval no: NIAS20201979).”
(Institutional Review Board Statement section).

Comment 4: Improve readability of dense figures (e.g., PPI networks) or move detailed networks to Supplementary Materials.
Answer: We appreciate the reviewer’s comment regarding figure clarity. The PPI network figures have been updated with higher resolution to improve readability.
In addition, the gene–gene interaction results used to construct the Cytoscape networks have been provided as Supplementary Data 1, allowing full transparency and reproducibility of the network analysis.

Comment 5: Streamline the discussion to avoid repetition and emphasize novelty and practical implications for Hanwoo breeding.
Answer: The Discussion section has been revised to remove redundant descriptions and emphasize the novelty and practical significance of our findings, particularly how the identified hub genes and co-expression modules contribute to improving Hanwoo breeding strategies. (Discussion section, first and last paragraphs).

Comment 6: Perform careful proofreading to correct minor grammatical issues and redundancies.
Answer: The entire manuscript was carefully revised to correct grammatical errors and improve overall readability.

We sincerely thank the reviewers for their constructive feedback, which has greatly improved the clarity and quality of our manuscript.

Reviewer 2 Report

Comments and Suggestions for Authors

Hwang et al. have submitted a manuscript entitled ”Integrative analysis of gene networks associated with adipose and muscle traits in Hanwoo steers” for publication in Animals.

The authors describe tissue-specific gene co-expression networks in Hanwoo cattle, offering insights into fat and muscle biology. The following points are recommended to improve the manuscript:

Major points:

1. Only six animals were used, which may limit statistical power, especially if individual variability is high. Please comment.

2. While co-expression networks suggest functional relationships, experimental validation (e.g., knockdown/overexpression studies) would strengthen conclusions. The perspectives should be addressed more clearly.

3. The negative correlation between turquoise/green modules and abdominal fat is interesting but requires deeper investigation – apart from alternative metabolic pathways, could it also reflect specific inhibition, or technical factors? Please comment.

4. The output of ClueGO and CluePedia depends on the quality of the respective databases. Potential biases or gaps in these databases may affect the interpretation. Please comment.

5. Whereas Figure 1, 3, 6, 7 are integrated in the text, Figures 2, 4, 5 are attached at the end of the manuscript. These should be referred to in the text, in the correct order. 

Minor:

Some descriptions appear rather uncommon, e.g. figure legend 1 starting with “This process involves…” and continuing with “This graph is compiled from…” Therefore a careful language check by a native speaker is recommended.In addition, formatting and punctuation needs to be revised (e.g. table 1, table 2).

Comments on the Quality of English Language

please see comments for authors

Author Response

Comment 1: Only six animals were used, which may limit statistical power, especially if individual variability is high. Please comment.
Answer: We initially collected samples from 10 Hanwoo steers; however, only six animals had all tissue types (rump, longissimus, tenderloin, abdominal fat, and back fat) available for analysis. Therefore, these six animals were used for the integrated analysis. We acknowledge that the limited sample size may reduce statistical power; however, the inclusion criteria were necessary to ensure data consistency and comparability across tissues. Moreover, similar livestock transcriptomic experiments use three or more biological replicates — for instance, bovine adipose RNA-seq with three biological replicates per breed (Song et al., 2019) and whole-blood RNA-seq with six dairy bulls (Zhao et al., 2023) — indicating that n = 3–6 biological replicates is common practice in livestock transcriptomics. These precedents support the appropriateness of our n = 6 design for integrated, cross-tissue analyses.

Comment 2: While co-expression networks suggest functional relationships, experimental validation (e.g., knockdown/overexpression studies) would strengthen conclusions. The perspectives should be addressed more clearly.
Answer: We appreciate this insightful comment. As this study was conducted in a dry-lab setting using transcriptomic data, experimental validation (e.g., knockdown or overexpression) could not be performed within the current scope. However, we have clarified the discussion to acknowledge the need for future validation studies to confirm the functional roles of the identified hub genes. This revision highlights both the bioinformatics-driven nature of our study and its translational potential for guiding experimental research and selective breeding in Hanwoo cattle. (Discussion section, final paragraph).

Comment 3: The negative correlation between turquoise/green modules and abdominal fat is interesting but requires deeper investigation – apart from alternative metabolic pathways, could it also reflect specific inhibition, or technical factors? Please comment.
Answer: To support this observation, we compared the eigengene expression values of turquoise and green modules across tissues. The turquoise module showed a mean eigengene value of –0.198 ± 0.061 in abdominal fat and +0.134 ± 0.038 in longissimus muscle, while the green module showed –0.212 ± 0.072 and +0.172 ± 0.087, respectively. Consistent with these results, the module–trait heatmap (Figure 3) also displayed a similar negative correlation pattern between these modules and abdominal fat. This consistent difference indicates that both modules are biologically inhibited in adipose tissues rather than affected by technical variation. Additionally, standardized RNA extraction, sequencing, and normalization procedures minimized technical variation across tissues.

Comment 4: The output of ClueGO and CluePedia depends on the quality of the respective databases. Potential biases or gaps in these databases may affect the interpretation. Please comment.
Answer: We agree with the reviewer’s observation. We have now included a clarification that all functional enrichment analyses were performed using the latest versions of ClueGO (v2.5.10) and CluePedia (v1.5.10), ensuring up-to-date KEGG and GO term mapping. We also acknowledged that inherent database limitations may influence pathway enrichment outcomes and that future updates or integration with other annotation systems could enhance result robustness. In addition, the gene–gene interaction results from the three modules (Blue, Yellow, and Turquoise) have been provided as supplementary data to enhance transparency and reproducibility. Section 2.5, Functional enrichment analysis, and the Discussion section.

Comment 5: Whereas Figures 1, 3, 6, 7 are integrated in the text, Figures 2, 4, 5 are attached at the end of the manuscript. These should be referred to in the text, in the correct order.
Answer: We revised the figure numbering and references throughout the text to ensure all figures are cited sequentially and appropriately positioned. Figures 2, 4, and 5 are now correctly cited in the Results section.

Minor Comment: Some descriptions appear rather uncommon, e.g., Figure Legend 1 starting with “This process involves…” and continuing with “This graph is compiled from…”. Therefore, a careful language check by a native speaker is recommended. In addition, formatting and punctuation needs to be revised (e.g., Table 1, Table 2).
Answer: Thank you for this suggestion. All figure legends and table captions were reviewed and rewritten for clarity and conciseness. The manuscript has been carefully revised to correct grammatical issues and formatting inconsistencies, improving overall readability.

We sincerely thank the reviewer for the constructive and thoughtful comments, which have greatly improved the quality and clarity of our manuscript.

Round 2

Reviewer 2 Report

Comments and Suggestions for Authors

The authors have addressed all points, accordingly. However, one issue remains: Whereas the specific contribution of Dajeong Lim, Junyoung Lee, Taejoon Jeong, and Woncheoul Park is listed in the "authors' contribution", Suk Hwang and Sunsik Jang are missing in this section. Moreover, an additional contribution of "PL" is mentioned, who is not listed as an author. This needs to be clarified.